# The Pharmacological TAILS of Matrix Metalloproteinases and Their Inhibitors

**DOI:** 10.3390/ph14010031

**Published:** 2020-12-31

**Authors:** Nabangshu Das, Colette Benko, Sean E. Gill, Antoine Dufour

**Affiliations:** 1Faculty of Kinesiology, University of Calgary, Calgary, AB T2N 4N1, Canada; nabangshu.das@ucalgary.ca; 2McCaig Institute for Bone and Join Healthy, 3280 Hospital Drive NW, Calgary, AB T2N 4Z6, Canada; cbenko3@uwo.ca; 3Department of Physiology and Pharmacology, Cumming School of Medicine, Foothills Hospital, 3330 Hospital Dr, Calgary, AB T2N 4N1, Canada; 4Centre for Critical Illness Research, Victoria Research Labs, Lawson Health Research Institute, A6-134, London, ON N6A 5W9, Canada; sgill8@uwo.ca; 5Division of Respirology, Department of Medicine, Western University, London, ON N6A 5W9, Canada

**Keywords:** matrix metalloproteinases (MMPs), protease, tissue inhibitors of metalloproteinases (TIMPs), exosite, small molecule inhibitors, monoclonal antibodies, proteomics, N-terminomics, terminal amine isotopic labeling of substrates (TAILS)

## Abstract

Matrix metalloproteinases (MMPs) have been demonstrated to have both detrimental and protective functions in inflammatory diseases. Several MMP inhibitors, with the exception of Periostat^®^, have failed in Phase III clinical trials. As an alternative strategy, recent efforts have been focussed on the development of more selective inhibitors or targeting other domains than their active sites through specific small molecule inhibitors or monoclonal antibodies. Here, we present some examples that aim to better understand the mechanisms of conformational changes/allosteric control of MMPs functions. In addition to MMP inhibitors, we discuss unbiased global approaches, such as proteomics and N-terminomics, to identify new MMP substrates. We present some examples of new MMP substrates and their implications in regulating biological functions. By characterizing the roles and substrates of individual MMP, MMP inhibitors could be utilized more effectively in the optimal disease context or in diseases never tested before where MMP activity is elevated and contributing to disease progression.

## 1. Introduction

Matrix metalloproteinases (MMPs) are zinc-dependent proteases that have been extensively studied in the context of extracellular matrix (ECM) breakdown and remodelling [1]. Increasingly, non-ECM substrates are being investigated for MMPs as ECM substrates only account for approximately 30% of all known MMP substrates [2,3]. The dysregulation of MMPs, their substrates, and the tissue inhibitor of metalloproteinases (TIMPs) often results in the progression of numerous diseases [1,3,4]. Various MMPs have been implicated in multiple cancers including pancreas, brain, lung, prostate, breast, skin and gastrointestinal tract [3,5]. MMP12 has been studied in chronic obstructive pulmonary disease (COPD) and the minor allele of a single nucleotide polymorphism in MMP12 (rs2276109) was associated with a beneficial effect on lung function in smokers and children with asthma [6,7]. Multiple MMPs have been investigated in rheumatoid arthritis and osteoarthritis yet the precise functions of individual MMP remains to be better characterized (reviewed in [8]). MMPs have also been studied in context of periodontal diseases [9,10]. It is not surprising that MMP inhibitors were tested in clinical trials. However, to date, the only MMP inhibitor that is currently approved is Periostat^®^ (doxycycline hyclate), which is used for treating periodontitis (Figure 1a). Despite their biological roles in multiple cancers, in addition to inflammatory and autoimmune diseases, most MMP inhibitors failed due to a combination of factors including poor study design, a lack of understanding of biological roles of MMPs and the substrates they cleave, and the lack of specific inhibitors [1,3,5,11]. The structures and amino acid sequence of the catalytic domain of the 23 MMPs are highly conserved, which initially resulted in the design of broad spectrum MMP inhibitors. MMPs, however, have both detrimental and protective functions, limiting the use of these broad-spectrum inhibitors and increasing the complexity of developing MMP inhibitors to treat human diseases. 

## 2. Regulation of MMP Activity

The catalytic activity of MMPs is tightly regulated by endogenous TIMPs [12]. TIMPs are secreted proteins that inhibit metalloproteinases [13] through the formation of 1:1 stoichiometric complexes [12]. The C-terminus of TIMPs interacts with the hemopexin like domain, found in all MMPs except MMP7 and MMP26, whereas the N-terminus interacts with the zinc ion within the catalytic domains of MMPs [14]. When an imbalance between MMPs and TIMPs occurs, it often results in inflammation and immune responses, as seen in many inflammatory diseases and cancers [15]. Therefore, the reestablishment of MMP-TIMP homeostasis is of pharmacological value and supports the need for the development of effective MMP inhibitors. Moreover, a better understanding of the biological functions of TIMPs is also needed to clarify their roles in human pathologies.

## 3. Non-Proteolytic Functions of MMPs

As demonstrated in previous clinical trials, broad spectrum targeting of the catalytic domain of MMPs is challenging. Thus, alternative methods for the inhibition of MMP functions have been investigated such as targeting exosites and ectosites. Not only would this potentially enable greater specificity between MMPs, but some exosites may have unique functions distinct from proteolysis. One example is the hemopexin (PEX) domain that contributes to protein-protein interactions and can initiate cell signalling and increased cell migration [16,17,18]. Since the amino acid sequences of the PEX domain across MMPs is more divergent and less conserved than the catalytic domain, the PEX domain is a potential site to target with inhibitors to increase selectivity. Interestingly, MMP7 and MMP26 do not contain a PEX domain, therefore not all MMPs require this domain. The MMP1 PEX domain is essential for binding to collagen and in the modulation of the triple helical structure of the substrate to allow access to the catalytic cleft [19]. Both the catalytic and PEX domain of MMP1 are necessary for the cooperative binding of triple helix collagen, demonstrating the importance of the PEX domain in substrate binding. Additionally, the conserved collagen residue P10 interacts with MMP1 via a hydrophobic pocket or exosite composed of Phe301, Ile271, and Arg27 within the PEX domain [19]. Further, when double mutants of Ile271Ala/Arg272Ala were generated, the collagenolytic function was significantly reduced. Thus, the inhibition of this hydrophobic pocket could potentially be a therapeutic approach to regulate MMP1 activity as it is important in not only the binding of triple helix collagen but in the processing of collagen [19]. The PEX domain of MMP12 plays a critical role in clearance of various bacteria such as *Staphylococcus aureus, Klebsiella pneumoniae*, *Escherichia coli*, and *Salmonella enteriditis* in the phagolysosome [18]. In *Mmp12^−/−^* mice, there was an increase in mortality at lower titer concentration when infected with *S. aureus* as compared with wild type mice [18]. Anti-bacterial properties of MMP12 were determined to be the result of disruption of the bacterial outer membrane by amino acids 344-363 in blade II of the PEX domain [18]. Conversely, the catalytic domain of MMP12 may contribute to the cleavage of bacterial toxins but did not demonstrate antibacterial properties against *S. aureus* α-toxins [18]. Therefore, a better characterization of the PEX domains of MMPs may reveal new exciting functions in other MMPs.

The PEX domain of MMPs is also implicated in homo-/hetero-dimerization and can form multimers [20]. The propeller structure of the PEX domain includes 4 blades composed of two alpha-helices and four beta strands [21]. In MMP9, a mutation in blade IV of the PEX domain resulted in a loss of homodimer formation [16]. Mutations in blade I of the MMP9 PEX domain resulted in a loss of interactions with the cell surface CD44 [16]. This interaction between the outer blade I of the MMP9 PEX domain and CD44 was shown to increase cell migration via the activation of epidermal growth factor receptor (EGFR) and downstream kinase signaling [16]. Peptides generated to mimic the outer beta strand of blade I or IV resulted in decreased levels of MMP9 dimers and also a reduction cell migration [16]. MMP9 can also increase angiogenesis [22]. Using an allosteric inhibitor to the PEX domain, Hariono et al. [22] demonstrated that inhibition of ECM proteolysis, which decreases the release of vascular endothelial growth factor (VEGF) from within the ECM, significantly reduces the binding of VEGF to its membrane receptor, and subsequently decreases angiogenesis.

The catalytic domain of membrane type 1-matrix metalloproteinase/MT1-MMP (MMP14) has been implicated in pro-tumorigenic functions by processing type I collagen, in addition to increasing cell migration, angiogenesis, and cell invasion [23,24,25]. The PEX domain of MT1-MMP also forms hetero- (with CD44) and homo-dimers via blades I and IV of the PEX domain, respectively [25]. Synthetic peptides mimicking the outermost strand motifs within the PEX domain (blades I and IV) of MT1-MMP were shown to specifically inhibit MT1-MMP-enhanced cell migration, although the ability to directly prevent MT1-MMP proteolytic activity was not shown [25]. The PEX domain contributes to the tumor promoting nature of MT1-MMP as tumour volume was significantly larger in cancer cells containing the PEX domain compared to those without [24]. MT1-MMP also contains transmembrane and cytoplasmic tail domains that have been shown to have distinct functions from the catalytic domain and could be targeted with inhibitors to interfere with the biological functions of MT1-MMP. Targeting the PEX domain of MMPs could provide non-competitive inhibition as compared with active site inhibition with broad-spectrum compounds [26]. Each MMP is likely to have unique exosites or “hotspots” that may be targeted individually due to divergence of their amino acid sequences, chemical potential and geometry [27]. However, the binding affinity of most exosites for substrate is typically low (10^−6^–10^−7^ M) making it potentially challenging to design an effective drug against that site [28,29].

## 4. Strategies for the Development of Protease Inhibitors

Multiple MMP inhibitors were originally designed with a substrate-based peptide, resembling the structure of type I collagen where MMPs cleave, aimed to interact with the necessary zinc ion in the MMPs’ active site [30]. This active site zinc ion is a required component of their catalytic site activity [31], coordinated by three histidine residues, and calcium ions, which stabilize conformation of the active protease [32]. Examples of chemical groups with zinc chelating agents used in the development of MMP inhibitors include hydroxamates, carboxylates, aminocarboxylates, phosphonate, and sulfhydryl groups [28,33]. Table 1, Table 2 and Table 3 provides a summary of MMP inhibitors that were tested in clinical trials and pre-clinical studies. One example is Batimastat, a peptidomimetic composed of a hydroxamate group, that was investigated for the treatment of breast cancer and was terminated in clinical trials due to its poor solubility and low oral bioavailability [34,35]. Later, a chemical analogue, marimastat, with improved oral bioavailability, was taken further along in clinical trials and was terminated due to musculoskeletal pain and lack of efficacy [34]. Another hydroxamate derivative, Prinomastat, was also unsuccessful in phase III clinical trials due to lack of efficacy in patients with late-stage disease [36]. While the use of MMP inhibitors in combination with traditional chemotherapy drugs was reported to improve adverse side effects, the chemotherapeutics, in turn, surprisingly lowered the therapeutic effects of the MMP inhibitors [37,38,39,40]. Despite termination of clinical trials for MMP inhibitors for the treatment of cancer and arthritis, another MMP inhibitor, Periostat^®^ (CollaGenex Pharmaceuticals Inc.), was successfully approved for the treatment of periodontitis [41,42,43]. Periostat^®^ is a synthetic tetracycline, (doxycycline hyclate) but its precise mechanism of action on MMPs activity remains unclarified [42]. Periostat^®^’s ability to bind to the calcified surfaces of tooth roots may potentiate its efficacy in periodontal disease [44]. The gradual release of doxycycline from teeth in active form also may contribute to increased exposure, the maintained effectiveness during the post-treatment period [42]. Periostat^®^ also reduces the level of localized and systemic inflammatory mediators in osteopenic patients in addition to improving on the clinical measurements of periodontitis [45]. Additionally, it has showed therapeutic effects in multiple sclerosis and type II diabetes. In multiple sclerosis (MS), in a combination therapy with intramuscular interferon-β (IFNβ), oral doxycycline was found to be effective, safe and well-tolerated [46]. In this study, outcome measures included number of lesion changes, relapse rates, safety and tolerability of the combination therapy in patients with MS. Multiple parameters were recorded including the Expanded Disability Status Scale scores, MMP9 levels in the serum, and the transendothelial migration of monocytes exposed to serum from patients with relapsing-remitting multiple sclerosis (RRMS). The inhibitory effect of doxycycline was associated with decreased serum level of MMP9 and was found to be corelated with reduction in brain lesion activity as measured by gadolinium-enhancing lesion number change [46]. When serum from RRMS patients was incubated with monocytes, their transendothelial migration was significantly diminished. Importantly, in this study, the adverse effects were mild, and one out of fifteen patients relapsed. In another clinical trial with obese people with type II diabetes, doxycycline was tested over a 12-week timepoint resulting in decreased inflammation and improved insulin sensitivity [47]. This effect was associated with a decrease in C-reactive protein and myeloperoxidase comparing to the placebo; it also increased 3′- phosphoinositide kinease-1, protein kinase B, and glycogen synthase kinase 3 ß [47]. However, these clinical trials were performed only on a small number of patients and further studies on a larger number of patients are needed to further test their efficacy. The failure of all MMP inhibitors in clinical trials, with the exception of Periostat^®^, led to the investigation of the roles of MMPs beyond their recognized roles in ECM remodeling [5]. Recent studies using animal model of disease coupled with high-throughput methods for substrate discovery (will be further discussed in Section 8) have revealed important roles of MMPs in inflammation and viral/bacterial infections [3,11,18,48,49]. In fact, multiple MMPs, such as MMP2, -3, -8, -9, and -12, play important roles in maintaining tissue homeostasis and have been demonstrated to have protective effects (full list is reviewed in [3]). Therefore, the ideal MMP inhibitor should be able to interfere with detrimental MMPs while sparing the beneficial MMPs. The catalytic domains of MMPs are highly conserved, therefore, targeting other MMP domains, which are unique to a single MMP, represents an alternative method of MMP inhibitor design.

## 5. Small Molecule MMP Inhibitors

Strategies to inhibit MMP functions with small molecules has been explored (Table 2) [16,22,24,25,50]. Italicize in silico analysis of MMP9 in which molecular docking programs were utilized to map potential ligand binding sites in the PEX domain at the dimerization interface were successful at identifying compound that interfered with MMP9 homodimerization and blocked a downstream signaling pathway critical for MMP9 mediated cell migration and invasion [50]. These compounds (Figure 1b–e) spared MMP9′s proteolytic activity, and ‘compound 2′ (Figure 1c) from their study significantly diminished the phosphorylation of extracellular signal-regulated kinase 1/2 (ERK1/2). In a xenograft model of metastatic breast cancer cells (MDA-MB-435) stably transfected with a green fluorescent protein (GFP), a small molecule exosite MMP9 inhibitor, ‘compound 2′, significantly decreased the tumor size and reduced the number of lung metastasis [50]. A follow-up study [17] demonstrated that treatment with their newly identified compound (Figure 1f) disrupted MMP9 homodimerization and prevented association with α4β1 integrin, CD44 and decreased phosphorylation of Src and its downstream target proteins focal adhesion kinase and paxillin. In the in vivo model of chick chorioallantoic membrane, treatment with a PEX domain small molecule inhibitor resulted in a reduction of cancer cell invasion and angiogenesis [17]. Using previously published active compounds (Figure 1c,f), Hariono et al. [22] evaluated ten additional arylamide compounds. Using molecular dynamic simulations, the mechanism of MMP inhibition via the hemopexin domain of MMP9 was investigated. Two compounds, (3-bromo-N-(4-nitrophenyl)propanamide) and (3-bromo-N- {4-[(pyrimidine-2-yl)sulfamoyl]phenyl}propanamide), demonstrated significant inhibition and cytotoxicity against 4T1 murine breast cancer cells. Using a biochemical and structural screening workflow, the compound JNJ0966 (Figure 1g) was found to selectively impede activation of proMMP9 into its active form via an interaction with a structural pocket in proximity to the MMP9 zymogen cleavage site near Arg106 [29]. JNJ0966 was unable to interact with the active forms of MMP1, -2, -3, -9, or -14. In a mouse experimental autoimmune encephalomyelitis model, JNJ0966 reduced disease severity at a dose of 10 and 30 mg/kg and to the same levels of dexamethasone [29]. Italicize in silico analysis of MT1-MMP’s PEX domain identified a potentially targetable site that is distinct from the dimerization interface and located at the center of the PEX domain [24]. Subsequent docking studies of a small molecule inhibitor led to identification of a novel PEX inhibitor which is selective for MT1-MMP as compared to MMP2. It did not show any toxicity or interference to catalytic activities including MT1-MMP mediated activation of MMP2. This compound was effective in attenuating cancer cell migration and reduced tumor volume in vivo [24]. Despite promising results, there are still only a few studies that have tested small molecule MMP inhibitors in animal models and additional work in characterizing these inhibitors is needed to better understand whether this could be a viable approach to inhibit MMP function.

## 6. MMP Inhibition Using Selective Antibody

Monoclonal antibodies have emerged as potential enzyme inhibitors with numerous examples demonstrating them to be effective [51,52,53,54]. Initially, antibody generation was deemed challenging for the active site of MMPs due to the instability during presentation and lack of surface accessibility of the catalytic metal-protein cleft [55]. Sela-Passwell et al. [55] used a synthetic metal-baring mimicry complex and were able to generate a response not only to the metal-protein cleft, but also to the enzyme surface of MMP2 and MMP9 with high specificity which resulted in the generation of the monoclonal antibody SDS3. In a dextran sodium sulfate (DSS)-colitis mouse model, SDS3 was demonstrated to prevent colonic inflammation, release of proinflammatory cytokines, and tissue damage [55]. The applicability of SDS3 towards colitis and inflammatory bowel disorders was reinforced through clustering of the labelled SDS3 antibody in the intestine of the mice 24 h after injection. Another MMP9 monoclonal antibody inhibitor, REGA-3G12, was demonstrated to be selective towards MMP9 through recognition of the N-amino terminal catalytic domain subsequently inhibiting its catalysis [56,57]. REGA-3G12 recognizes the Trp116-Lys214 motif domain part of the Phe107-Gly223 outside of the Zn^2+^ binding site of MMP9, a site exploited by multiple MMP inhibitors. Although REGA-3G12 targets the catalytic domain, it was shown to be selective to MMP9 but not MMP2, therefore demonstrating the potential of the catalytic domain to be used to generate selective MMP inhibitors. The selectivity of REGA-3G12 is likely due to the vast number of interactions with the carboxy-terminal of the catalytic domain. REGA-3G12 did not show significant binding to synthetic linear fragments of the epitopes recognized by REGA-3G12, demonstrating the antibody may recognize a conformation instead of a linear residue. MT1-MMP inhibition is another attractive anti-cancer target as it is highly expressed in breast cancer and contributes to migration, invasion, and neovascularization [58]. Using a phage library display approach, an MT1-MMP active site inhibitor antibody, DX-2400, was developed to selectively inhibit active MT1-MMP. DX-2400 was also shown to inhibit pro-MMP2 activation due to the inhibition of MT1-MMP and reduced breast cancer cell invasion [58]. DX-2400 exhibited promising therapeutic results in an MDA-MB-231 breast cancer xenograft tumor mouse model associated with a decrease in cell growth and vascularization. Conversely, non-metastatic MCF-7 breast cancer cell that do not express MT1-MMP, did not show decreased growth in a xenograft tumor mouse model [58]. DX-2400 also delayed metastasis, and when in combination with bevacizumab delayed breast cancer cells’ tumor growth [58]. Similarly, when applied to a BT-474 xenograft tumor mouse model, DX-2400 in combination with paclitaxel resulted in tumour growth delay. Surprisingly, while this study demonstrated promising results related to using selective MT1-MMP inhibitors to treat breast cancer, DX-2400 was unsuccessful in clinical trials.

Selective inhibition of MT1-MMP was also demonstrated in a model of influenza infections that resulted in ECM dysregulation and increased susceptibility for a bacterial co-infection [48]. The selective, potent and allosteric MT1-MMP inhibitor, LEM-2/15, was generated via mouse immunization with a cyclic peptide from the sequence of the V-B loop (residues 218–233), which displays a unique sequence divergence within the MMP family members [59]. This loop of MT1-MMP is flexible and likely undergoes conformational changes when binding to LEM-2/15 causing a narrower substrate binding cleft and constraining the flexibility of the loop. Therefore, the Fab fragment of LEM-2/15 interacted with the MT1-MMP expressed on the cell surface and inhibited its collagenase activity while not interfering significantly with the activation of proMMP2 and MT1-MMP homodimerization on the cell surface [59]. ECM remodeling is usually independent of the viral burden but rather linked to the proteolysis driven by the immune response. LEM-2/15 reduced inflammation and ECM remodeling during influenza infection [48]. When used prophylactically or therapeutically during a coinfection of influenza and *S. pneumoniae*, LEM-2/15 significantly improved survival in mice. Interestingly, when used in combination with Tamiflu^®^ (Oseltamivir) both prophylactically or therapeutically, an approved anti-influenza inhibitor, 100% survival was achieved in a mouse model of viral infection [59]. Importantly, Tamiflu^®^ alone resulted in increased survival only when used prophylactically. Thus, promoting ECM stability and homeostasis via MT1-MMP inhibition during influenza infection is an attractive target.

## 7. MMP Substrates Extend Beyond Matrix Proteins

Contrary to what their name suggests, MMPs have been shown to cleave substrates other than matrix (ECM) proteins [1,2,11]. MMPs cleave chemokines and cytokines to regulate their functions [49,60,61,62,63]. For example, the processing of monocyte chemoattractant proteins, CCL-7 and CCL-13, reduced the inflammatory response, as demonstrated in mouse model of inflammatory edema [64]. In a mouse model of asthma, MMP2 and MMP9 were found to be protective via disruption of transepithelial chemokine gradients regulated by CCL7, CCL11, and CCL17 [65]. In macrophages, MMP12 cleaved the C-terminus of IFNγ, removing the receptor binding site and thereby decreasing JAK–STAT1 signaling and IFNγ activation within the proinflammatory macrophage [63]. Genetic ablation of MMP12 or therapeutic inhibition of MMP12 using Rxp470.1 in murine models of autoimmune inflammatory diseases resulted in elevated IFNγ mediated inflammatory signatures compared to the control groups [63]. In virus-infected cells, MMP12 was shown to be localized in the nucleus and promoted *NFKBIA* transcriptional activity resulting in INFα secretion, a key mechanisms for antiviral immunity [49]. In parallel, extracellular MMP12 attenuated systemic IFNα, and use of an MMP12 inhibitor, Rxp470.1, which is unable to enter the cells, significantly reduced the viral load [49]. MMP processing of CCL15 and CCL23, implicated in inflammatory arthritis, resulted in an increase in monocyte recruitment during inflammation [62]. Collectively, these four lines of evidence support the critical role for MMPs in regulating the inflammatory response through direct cleavage of chemokines/cytokines. Further, only 30% of MMP substrates are linked with the ECM [2]. Therefore, additional roles of MMPs are likely to be identified when looking beyond the matrix.

## 8. Identification of Novel MMP Substrates Using N-Terminomics/TAILS

N-terminomics technologies have been used to profile and identify new MMP substrates in various cell systems and tissues [2,3,49,66,67]. One example is terminal amine isotopic labeling of substrates (TAILS), a high throughput quantitative proteomic platform that allows simultaneous quantitative analysis of the N-terminome and proteolysis on a proteome-wide scale, and hence allows for protease substrate discovery [68]. To study the substrate repertoire of a specific protease using TAILS, the protease of interest can be compared to an inactivated form of the protease or to a protease inhibitor-treated sample (Figure 2). Alternatively, tissue proteomes of protease knock-out mice can be compared to wild-type animals with or without induction of a specific infection, stress or disease. Once collected, proteomes are denatured, and it is important to avoid primary amine containing buffers. After the reduction and alkylation of cysteine residues, primary amines of both the N-termini and lysine residues are chemically labeled with formaldehyde. During this step, we incorporate stable isotope labeling in order to later compare the different conditions being tested. An example of isotopic labeling is light (+28 Da) and heavy (+34 Da) dimethylation used with the catalyst sodium cyanoborohydride (NaBH3CN) [69]. These isotopic modification can later be monitored using liquid chromatography and tandem mass spectrometry (LC-MS/MS) [68]. However, dimethylation reactions are limited to three distinct labels and other labelling such tandem mass tag (TMT) can label up to 11 different samples with a distinct isotope [67,70]. After isotopic labeling, the labeled proteomes are then mixed and digested with trypsin. During digestion, trypsin cleaves the peptide only after arginine (semi-ArgC specificity) as the blocked lysine residues are unreactive to trypsin. After trypsin cleavage, ~10% of the sample is collected and prepared for LC-MS/MS analysis; this is the pre-enrichment TAILS samples. Next, the TAILS aldehyde reactive polymer is used to remove the internal tryptic peptides that were generated during trypsin digestion. The unbound blocked and labeled peptides are recovered from the samples by size exclusion (10-kDa cut-off filters) filtration. The recovered peptides are then analyzed via LC-MS/MS analysis. The abundance ratio of blocked peptides from the TAILS samples can be compared to naturally blocked N-termini from the pre-enrichment TAILS samples. The neo-N-termini peptides specific to the protease of interest appear in higher ratios or only in the protease-treated sample and therefore show high protease/control abundance ratio. Therefore, the TAILS protocol is designed to identify new protease substrates and also identifies the precise cleavage site within the substrate sequence. TAILS has been used to profile the substrate repertoires of various MMPs and hundreds of new substrates have been identified in specific cell lines and tissues [66,71,72,73,74,75]. Using TAILS, the substrates of MMP2 and MMP9 were investigated in fibroblasts secretomes where 201 substrates were identified for MMP2 and only 19 for MMP9 [75]. Although, more MMP2 substrates were identified, most substrates can be cleaved by both MMP2 and MMP9 including thrombospondin-2, galectin-1, insulin-like growth factor-binding protein 4 (IGFBP4), dickkopf-related protein-3, and pyruvate kinase M1/M2 [75]. This study suggests that the regulation of MMP2 and MMP9’s activity could be linked with differences in genetic expression, different rates of TIMP inhibition, alternate activation mechanisms and/or distinct kinetic activities that could explain the differences in the phenotypes of *Mmp2^−/−^* and *Mmp9^−/−^* mice. Additional italicize in vivo studies comparing various cells, tissues, and organs of WT counterparts, *Mmp2^−/−^* and *Mmp9^−/−^* mice using disease models will help better characterize the unique roles and substrates of these two MMPs.

The substrates of macrophage MMP12 were investigated using TAILS by incubating murine MMP12 with secretomes from *Mmp12^−/−^* murine embryonic fibroblasts (MEFs), murine macrophage cell line RAW264.7 secretomes and also by comparing WT and *Mmp12^−/−^* peritoneal macrophages from a peritonitis model using thioglycollate stimulation for 4 days [71]. Hundreds of MMP12 substrates were identified including pyruvate kinase, biglycan, vimentin, renin-receptor and alpha-2-HS-glycoprotein (see [71] for the full list of substrates). Using TAILS, new functions for MMP12 in coagulation, complement activation/deactivation and resolution of inflammation were identified. To further confirm these MMP12 substrates in human diseases, TAILS was used to investigate nine COPD patients at exacerbation and recovery [7]. When comparing MMP12 substrates from murine peritonitis and joint inflammation model to the sputum of COPD patients, multiple identical substrates were identified including alpha-2-HS-glycoprotein, complement C3 (C3), complement C4-B (C4b), hemopexin, antithrombin III (SERPINC1), but also new substrates were identified such as transmembrane protease serine 7 (TMPRSS7) and DEP domain-containing mTOR-interacting protein (DEPTOR) [7,71]. MMP12 can cleave hundreds of substrates, therefore, the regulation of its activity is likely driven and impacted by other proteases, TIMPs, tissue specific microenvironment and other immune cells present such as neutrophils, eosinophils, natural killer cells, mast cells, T and B cells. Interestingly, MMP12 has been predominantly identified as a beneficial/protective MMP in inflammatory diseases although it has been implicated as a potential drug target in certain cancers. However, its precise role in inflammation needs to be further characterized [3,49,63,71].

Using TAILS, numerous non-ECM substrates have been identified further demonstrating new roles for MMPs (reviewed in [1,2,76]). For example, 58 new substrates were identified when MT6-MMP (MMP25) was added to fibroblasts secretomes, including vimentin, cystatin C, galectin-1, secreted protein acidic and rich in cysteine (SPARC), and insulin-like growth factor-binding protein 7 (IGFBP7) [72]. These identified substrates indicated a novel role for MT6-MMP for the clearance of apoptotic neutrophils. Cleavage of vimentin by MT6-MMP resulted in a decrease in chemoattraction of THP-1 monocytic cells but an increase in phagocytosis activity in an assay where fluorescent microbeads were coated with vimentin or cleaved vimentin and added to THP-1 cells [72]. The identification of new MT6-MMP substrates using TAILS supported a key biological role for this MMP in innate immunity and resolution of inflammation. Most MMPs have yet to be investigated using unbiased N-terminomics approaches to identify the extent of their substrates. It is anticipated that the cell type producing MMPs and specific microenvironments where MMPs are present will greatly impact what substrates can be cleaved by MMPs.

## 9. Conclusions and Perspective: Next Generation of MMP Inhibitors

MMPs have been shown to play a key biological role in numerous pathologies, however, the broad-spectrum inhibitors targeting multiple MMPs has potentially impeded their therapeutic applicability and use in the clinic to treat joint diseases or cancers. There are various pharmacological approaches that have been utilized to inhibit both the proteolytic and non-proteolytic functions of MMPs: peptides, monoclonal antibodies, and small molecule inhibitors (Figure 1 and Figure 3, Table 1, Table 2 and Table 3). With the discovery and development of novel specific methods of targeting individual MMPs, there is renewed hope for other MMP inhibitors and a revival in better characterizing MMP functions. Moreover, additional opportunities for the use of MMP inhibitors may become apparent. For example, given the success of MMP inhibition in animal models, MMP inhibitors could play an emerging role in veterinary medicine. For example, MMP2 and MMP9 inhibitors may be beneficial for the treatment of canine chronic enteropathy [77]. Elevated levels of active MMP2 and MMP9 are found throughout the intestines of dogs with chronic enteropathy and are associated with increased inflammation and neutrophil infiltration. MMP2/9 inhibitors could be effective in lowering intestinal inflammation and decreased disease severity [77]. Another example is the use of MMP inhibitors to treat the elevated expression of MMP2 and MT1-MMP in the narrowing of myocardial vessels in canine myxomatous mitral valve disease [78]. When considering application to human health, one option could be to only inhibit MMPs in diseases requiring short-term treatments, therefore limiting the period over which the treatment is received effectively minimizing any detrimental consequences that could arise. For example, sepsis could be further investigated for the short-term implementation of MMP inhibitors. In sepsis, MMPs cleave and regulate cytokine storm and chemokine activity, thus, playing a role in many of the pathways resulting in complications [79]. Another example is MMP12 with its anti-bacterial functions and its involvement in the inflammatory pathway leading to reduced lethality of lipopolysaccharide (LPS) induced inflammation [18,80]. Selective MT1-MMP inhibitors alone or in combination with Tamiflu^®^ also demonstrated efficacy in influenza infection models and could be further investigated for other viral infections. The utilization of shorter dosing and altering administration may reduce patient exposure to previously identified side effects of MMP inhibitors [28]. A more complete investigation into the protective and detrimental properties of MMPs and the subsequent development of therapeutics with high affinity for distinct MMPs may provide greater applications of MMP inhibition in the future.

## Figures and Tables

**Figure 1 pharmaceuticals-14-00031-f001:**
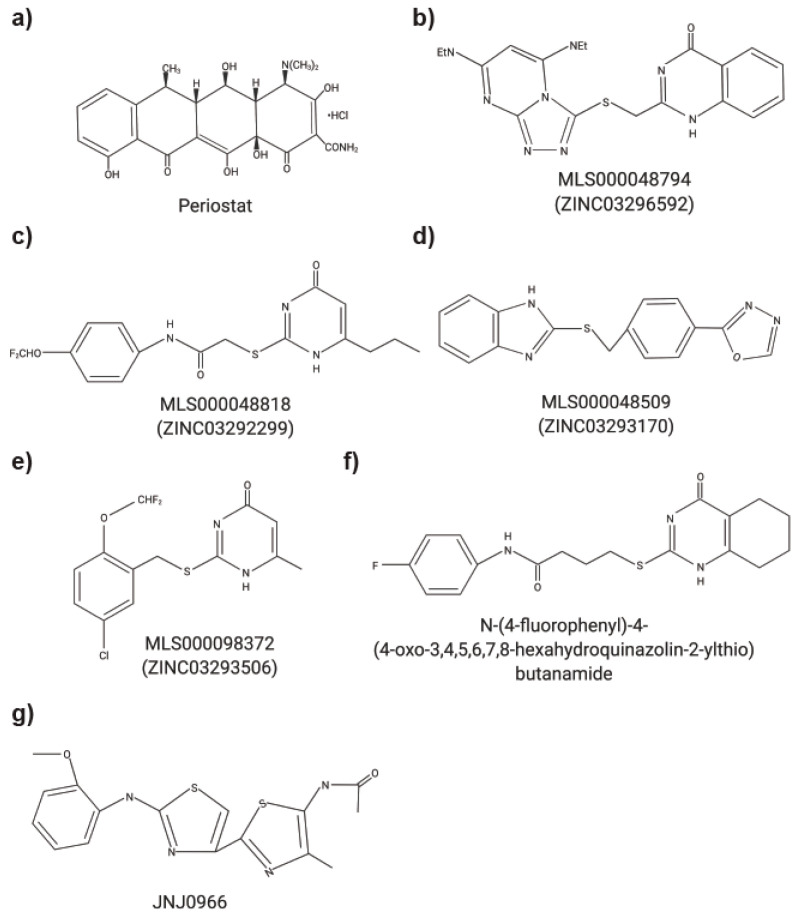
Chemical structures of small molecule MMP inhibitors. (**a**) Periostat^®^: doxycycline hyclate. (**b**) MLS000048794: 2-[[3,5bis(ethylamino)-2,4,6,8,9-pentazabicyclo[4 .3.0]nona-2,4,7,9-tetraen-7yl] sulfanyl-methyl]-1H-quinazolin4-one). (**c**) MLS000048818: *N*-[4-(difluoromethoxy)phenyl]-2-[(4-oxo-6-propyl-1H-pyrimidin-2-yl)sulfanyl]-acetamide. (**d**) MLS000048509: 2-[[4-(1,3,4-oxadiazol-2-yl)phenyl] methylsulfanyl]-1Hbenzoimidazole). (**e**) MLS000098372: 2-[[5-chloro-2 (difluoromethoxy)phenyl]methylsulfanyl]-6-methyl-1H-pyrimidin-4-one). (**f**) N-(4-fluorophenyl)-4-(4-oxo-3,4,5,6,7,8-hexahydroquinazolin-2-ylthio)butanamide. (**g**) JNJ0966: *N*-{2-[(2-methoxyphenyl)amino]-4′-methyl-4,5′-bi-1,3-thiazol-2′-yl}acetamide.

**Figure 2 pharmaceuticals-14-00031-f002:**
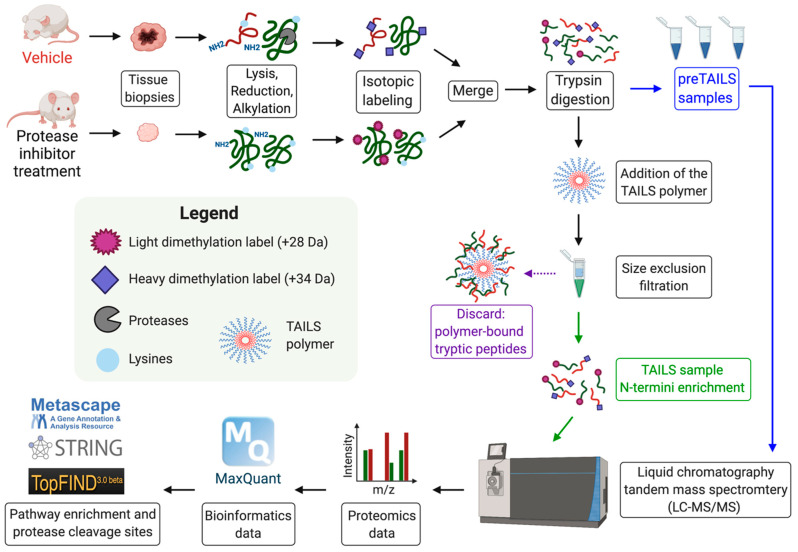
Experimental workflow of the N-terminomics/TAILS protocol using dimethylation isotopic labeling in a cancer mouse model treated with a protease inhibitor or a vehicle control. Tumors are lysed, reduced and alkylated before being isotopically labeled with light (+28 Da) or deuterated (+34 Da) formaldehyde. Proteins are then digested with trypsin and a pre-enrichment TAILS sample is collected. The remaining peptides are subjected to the TAILS polymer and N-termini are added to a size exclusion filter. Samples are subjected to liquid chromatography and tandem mass spectrometry (LC-MS/MS). The proteomics data is analyzed by bioinformatics software (e.g., MaxQuant). Data interpretation is analyzed with pathway enrichment tools (Metascape, STRING) or protease analysis software (TopFIND).

**Figure 3 pharmaceuticals-14-00031-f003:**
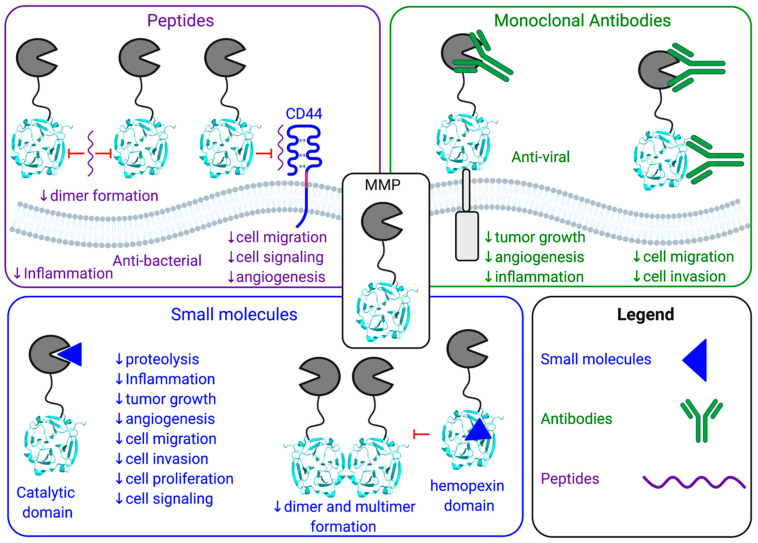
Schematic of three pharmacological approaches to inhibit the biological functions of MMPs: peptides (purple), monoclonal antibodies (green) and small molecules (blue).

**Table 1 pharmaceuticals-14-00031-t001:** MMPs inhibitors in clinical trials.

Inhibitor Names	Class/Structure	Selectivity	Diseases	Clinical Trial	Outcomes/ Side Effects	References
Batimastat	Peptidomimetic/ Hydroxamate	Broad spectrum	Malignant tumor	Phase I	Local toxicities i.e., abdominal discomfort	[81]
Marimastat	Peptidomimetic/ Hydroxamate	Broad spectrum	Progressive ovarian, prostatic, pancreatic and colorectal cancer	Phase III	Adverse musculoskeletal (MS) syndrome	[82]
			Pancreatic cancer	Phase III	Musculoskeletal pain and inflammation	[83]
			Pancreatic cancer	Phase III, in combination with gemcitabine	Well tolerated but no therapeutic beneficial effects	[40]
			Gastric cancer	Phase III	Severe musculoskeletal (MS) syndrome	[84]
			Metastatic breast cancer	Phase I	Musculoskeletal pain associated with inferior survival	[85]
MMI-270	Hydroxamate/Small molecule	Broad spectrum	Advanced solid cancer	Phase I	Rash and musculoskeletal pain	[86]
Prinomastat	Hydroxamate/Small molecule	Broad spectrum	Advanced cancer	Phase I	No response in tumor growth	[36]
			Non-small cell lung cancer (stage IIIB or IV)	Phase III	Musculoskeletal syndrome	[87]
			Esophagus cancer	Phase II	Unexpected thromboembolic events	[88]
Tanomastat (Bay 12-9566)	Biphenyl, thioether zinc-binding group/small molecule	MMP2, -3 and -9	Solid tumor	Phase I	Mild toxicity, no musculoskeletal pain. No effect on tumor	[89]
			Pancreatic cancer without prior chemotherapy	Phase III	Poorer survival	[90]
			Ovarian cancer	Phase III	Well tolerated but did not impact patients’ survival	[91]
Metastat	Tetracycline derivatives/small molecule	MMP2 and -9	Refractory solid tumors	Phase I	Subcutaneous phototoxicity	[92]
			AIDS related Kaposi’s sarcoma	Phase I Applied with sun protection	Photosensitivity reaction	[93]
			Advanced soft tissue sarcoma	Phase II Applied with sun protection	Photosensitivity reaction	[94]
Periostat^®^/ Doxycycline	Tetracycline derivatives/small molecule	Broad spectrum	Periodontitis	Phase III	Well tolerated; improved outcome	[41,43]
			Asymptomatic abdominal aortic aneurysms	Phase II	Well tolerated; but no significant therapeutic effects	[95]
			Multiple sclerosis	Phase II Along with IFNß-1a	Well tolerated; improved outcome	[46]
			Type II diabetes	Phase III	Reduced inflammation and better insulin sensitivity	[47]
Rebimastat (BMS- 275291)	Mercaptoacyl, thiol zinc-binding group/small molecule	MMP1, -2, -8, and MT1-MMP	Advanced cancer	Phase I	Well tolerated; no tumor response	[96]
			Early-stage breast cancer	Phase III	Study was terminated because of toxicity	[97]
			Non-small cell lung cancer	Phase II along with paclitaxel and carboplatin	Well tolerated but poor therapeutic response	[38]
			Non-small cell lung cancer	Phase II along with paclitaxel and carboplatin	Increased toxicity with no improved survival	[37]
S-3304	Sulfonamide derivatives/small molecule	MMP2 and -9	Advanced solid tumors	Phase I	Well tolerated	[98]
AZD1236		MMP9 and -12	Moderate to severe Chronic obstructive pulmonary disease (COPD)	Phase II	Well tolerated but no therapeutic efficacy	[99,100]
Neovastat (AE-941)	Mixed extract from shark cartilage	Broad spectrum	Non-small cell lung cancer (stage III)	Phase III along with chemotherapy	Well tolerated but no therapeutic effect	[39]

**Table 2 pharmaceuticals-14-00031-t002:** Small molecule exosite MMP inhibitors.

Compounds	Target	Binding Site	Mechanism of Action	Assays and Models Tested on	References
*N*-[4 (difluoromethoxy) phenyl] 2-[(4-oxo-6-propyl 1Hpyrimidin-2yl) sulfanyl]-acetamide	MMP9	Hemopexin (PEX)	Interfered with homodimerization; inhibition of cell migration and proliferation	Tumor growth and metastasis (xenograft mouse model)	[16]
NSC405020	MT1-MMP	Hemopexin (PEX)	Interfered with homodimerization and interaction with catalytic domain	Tumor growth (xenograft mouse model)	[24]
JNJ0966: (*N*-{2-[(2-methoxyphenyl) amino]-4-methyl-4,5-bi-1,3-thiazol-2-yl} acetamide)	MMP9	Pro-peptide domain	Inhibit activation of MMP9 without affecting MMP1, -2, -3, -9 and MT1-MMP	Autoimmune encephalomyelitis (mouse model)	[29]
*N*-(4-fluorophenyl)-4-(4-oxo-3,4,5,6,7,8-hexahydroquinazolin-2-ylthio) butanamide	MMP9	Hemopexin (PEX)	Inhibition of homodimerization; decreased cancer cell migration- blocks cancer cell invasion of basement membrane and angiogenesis	in vitro migration assay, Tumor growth (xenograft mouse model), angiogenesis (chicken chorioallantoic membrane)	[17]
Synthesized acrylamides (1) 3-bromo-*N*-(4-nitrophenyl) propenamide (2) 3-bromo-*N*-{4-[(pyrimidine-2-yl) sulfamoyl] phenyl} propenamide (3) 3-bromo-*N*-{4-[(4,6 dimethylpyrimidin-2-yl) sulfamoyl]-phenyl} propanamide	MMP9	Hemopexin (PEX)	Inhibition of 4T1 breast cancer cell growth; inhibition of MMP9 gelatinolytic activity	in vitro migration assay and (xenograft mouse model)	[22]

**Table 3 pharmaceuticals-14-00031-t003:** Regulation of MMP inhibition by antibodies.

Name	Antibody Type	Target	Epitope/Domains	Assays and Models Tested on	References
LEM-2/5	Monoclonal	MT1-MMP	Surface epitope; V-P loop	Migrating cancer cells;	[59]
				lung pathology; influenza	[48]
SD3	Monoclonal	MMP2 and -9	Catalytic domain	Inflammatory bowel disease (mouse model); colitis	[55]
REGA-3G12	Monoclonal	MMP-9	Catalytic domain other than Zn^2+^ binding	Inhibited MMP9 proteolytic activity	[56,57]
DX-2400	Fab fragment	MT1-MMP	Catalytic domain	Breast cancer	[58]
Multiple (A4-7 Fc-ScFv, E2_C6 Fc-ScFv)	Antibody fragments	MT1-MMP	Catalytic domain outside the active site cleft, inhibiting binding to triple helical collagen	Tumor growth and proliferation (xenograft mouse model)	[101]

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
