# Peer review of "The Pharmacological TAILS of Matrix Metalloproteinases and Their Inhibitors"

_pharmaceuticals, 2020, doi:10.3390/ph14010031_

Round 1

Reviewer 1 Report

Pharmaceuticals-1046228                                      Pharmaceuticals

The pharmacological TAILS of matrix metalloproteinases and their inhibitors

Matrix metalloproteinases (MMPs) belong to a large family of multidomain zinc endopeptidases. They  take part in digestion of components of the extra-cellular matrix but also in   many physiological processes, such as apoptosis, migration  or angiogenesis. MMPs are also engaged in the pathogenesis of many diseases such as arthritis, inflammation  or cancer. The development of effective inhibitors and discovery of their mechanisms of action can have significant influence on therapeutic strategy. The current paper is very interesting and collects information concernig brand new methods for the inhibition of MMP functions,  based mainly   on targeting exosites and ectosites or usage of selective monoclonal antibodies. The authors  present  also  MMP proteins as involved in many diseases.

Comments to authors on the article

Keywords:

  • TIMPs and TAILS should be explained here

Introduction:

  • some more information about involvement of MMPs in diseases should be presented here  to provide the reader with wider background  information
  • the abbreviation TIMPs  is twice  explained- see line 32 and 44

Chapter 4:

  • Line 117-119, 123: italic is needed for Latin names of bacteria mentioned here

Chapter 5:

  • Wider description is needed because the table 2 shows more examples but authors describe two of them only and keeping in mind that it is  the one of the main topic of this article

Chapter 8

-Similarly-it is very interesting part of this manuscript but also should be wider: some substrates must be presented here

Author Response

Dear Drs. Salvatore Santamaria and Amber Zhao,

      Thank you for handling our manuscript and your invitation to revise and resubmit. We are pleased that the reviewers felt that our review presented some recent findings and studies on MMP inhibitors. The reviewers suggested useful changes and additional information. Therefore, we performed a thorough re-write and have addressed all the points raised by the reviewers to the best of our ability. We thank the reviewers for their excellent insights and constructive comments that have led to a much-improved manuscript.

Review Report 1

Keywords:

TIMPs and TAILS should be explained here.

Agreed. Keywords have been explained.

Introduction:

Some more information about involvement of MMPs in diseases should be presented here to provide the reader with wider background information

Agreed and this information has been added to the introduction.

the abbreviation TIMPs is twice explained- see line 32 and 44

The second explanation has been removed.

Chapter 4: Line 117-119, 123: italic is needed for Latin names of bacteria mentioned here

We agree with the reviewer and it was italicized in our original manuscript. We think it is a PDF conversion issue. We have now checked and the names of all bacteria are italicized.

Chapter 5: Wider description is needed because the table 2 shows more examples but authors describe two of them only and keeping in mind that it is the one of the main topic of this article.

We agree with the reviewer. As there are only 5 studies published to our knowledge, we have discussed all the mentioned studies in chapter 5 and Table 2. Importantly, reference #96 has been moved to the antibody section as they have tested small antibody sequences and not small synthetic MMP inhibitors. Additional information has also been added to chapter 5.

Chapter 8: Similarly-it is very interesting part of this manuscript but also should be wider: some substrates must be presented here

Agreed. Additional information and papers describing MMP substrates have been added.

Reviewer 2 Report

The review by Nabangshu Das et al. describes the recent efforts and strategies to develop selective MMPs inhibitors. They Focus on three strategies: peptides, small molecules and monoclonal antibodies used to regulate MMPs functioning. They also describe N-terminomics/TAILS protocol for identification of novel substrates of these proteases. Even though the subject of MMPs inhibitors has already been taken couple of times (e.g. Winer et al. 2018, Mol Cancer Ther, Levin et al. 2017 Biochimica et Biophysica Acta (BBA) - Molecular Cell Research, Hu et al. 2005 Nature Reviews Drug Discovery) the review presented by Das et al. presents very recent knowledge in this area and describes new approaches and future perspectives regarding selective inhibition of MMPs.

In general, manuscript is written in very good English, and usually is easy to follow, however I have a feeling that sometimes authors wanted to give too much information in one section. Maybe it would be possible to divide sections into subsections?

I also suggest that section 4. Non-proteolytic functions of MMPs  should be moved after section 2. Regulation of MMP activity. I would also expect clearer justification in abstract and introduction for including MMPs substrates in this manuscript.

What is more, I think it would be valuable to present the structure of Periostat. Are the authors able to explain why Periostat was the only one successful in the clinical trials ? What was the difference in compare to e.g. prinomastat and others ?

The review is lacking conclusion. I would expect also authors’ critical opinion on what has been done in this subject up to date as well as on future perspectives.

Also the following needs to be corrected/explained:

It looks like there are double spaces between words in many places (reviewer has only access to PDF so it is hard to say that for sure), e.g. line 63 after chemical, line 79 after interferon and IFN, line 228 after Tamiflu, line 230 after Tamiflu, line 245 after IFN, etc.

Line 37 it should be domain not domain

Line 44 The abbreviation TIMPS was already explained in previous paragraph

Line 49 it should be cancers not cancer

Line 54 authors should provide the sequence of the mentioned peptide backbone 11

Lines 80-82 Sentence: This effect was associated with decreased serum level of MMP9 was found to be corelated with reduction in brain lesion activity as measured by gadolinium-  enhancing lesion number change [28].
is not clear. Please rephrase.

Line 85 should be 3’-phosphoinositide kinease-1 (missed dash)

Line 87 smaller than what ?

Line 89 explanation for ECM abbreviation was given in introduction

Lines 93-95 Authors should clarify or give examples which MMPs are considered as beneficial and which are not.

Line 101 should be investigated not investigating

Lines 133-134 MMP9 or MMP-9 ? please unite

Line 153 powers should be in superscripts

Line 160 what compounds ? can authors provide structures?

Line 169 can authors provide the structure of JNJ0966 ?

Line 189 is SDS3 the antibody inhibitor ? Please clarify and rephrase the sentence.

Line 196 should be Zn2+ (superscript)

Line 203 What cancer ? please specify.

Line 206 Again, what cancer ?

Lines 210-214 It also relates to breast cancer mentioned in line 209 ?

Line 287-288 Please develop this sentence “however, the broad-spectrum targeting has impeded their therapeutic applicability”

Figure 1 needs more detailed explanation in caption

Author Response

Dear Drs. Salvatore Santamaria and Amber Zhao,

      Thank you for handling our manuscript and your invitation to revise and resubmit. We are pleased that the reviewers felt that our review presented some recent findings and studies on MMP inhibitors. The reviewers suggested useful changes and additional information. Therefore, we performed a thorough re-write and have addressed all the points raised by the reviewers to the best of our ability. We thank the reviewers for their excellent insights and constructive comments that have led to a much-improved manuscript.

Review Report 2

In general, manuscript is written in very good English, and usually is easy to follow, however I have a feeling that sometimes authors wanted to give too much information in one section.

Maybe it would be possible to divide sections into subsections?

We have done significant rewrite so we hope that the reviewer finds the review easier to read.

I also suggest that section 4. Non-proteolytic functions of MMPs should be moved after section 2. Regulation of MMP activity.

Done as suggested.

I would also expect clearer justification in abstract and introduction for including MMPs substrates in this manuscript.

Done as suggested.

What is more, I think it would be valuable to present the structure of Periostat.

We have added a NEW figure (Figure 1a) where we present the structure of Periostat and other small molecule MMP inhibitors described in our manuscript.

Are the authors able to explain why Periostat was the only one successful in the clinical trials ?

What was the difference in compare to e.g. prinomastat and others ?

We now mention in the manuscript that it is likely that the microenvironment is dictating what MMP substrates are cleaved. Targeting periodontits is likely a beneficial factor of why Periostat was approved but no evidence can support this, unfortunately. It is only speculation at this point. In fact, it is hard to prove the precise reason why Periostat worked and not other MMP inhibitors as these drugs were tested in multiple diseases. Some studies mention that the reason Periostat was effective is because it can inhibit bacterial collagenases (not only MMPs) and diminish the total bacterial protease activity. Periostat impact bacterial functions and could be contributing to improvement of periodontits patients but no studies exist that support these claims. It has not been demonstrated in clinical trials so we decided to avoid discussing this point as not enough evidence exist.

We cite multiple reviews and articles that discuss the difference between various MMP inhibitors throughtout our manuscript. It is hard to comment on why some drugs were more or less effective as multiple diseases have been subjected to various MMP inhibitors and the trials were designed in different ways and tested in different populations. This is likely multi-factorial and is challenging to present an argument supporting our claims when no studies present clinical trials of the same MMP inhibitors across different diseases.

The review is lacking conclusion. I would expect also authors’ critical opinion on what has been done in this subject up to date as well as on future perspectives.

Chapter 9 has been updated and is now a conclusion section. We present novel perspective and directions of the next generation of MMP inhibitors. For example, using MMP inhibitor to treat viral infections or bacterial infections (sepsis). Also, MMP inhibitors could be used in veterinary medicine as it could be approved more rapidly than in human diseases. All these applications have not been tested yet in clinical trials and, we think, it could be attractive directions to go in the future.

Also the following needs to be corrected/explained: It looks like there are double spaces between words in many places (reviewer has only access to PDF so it is hard to say that for sure), e.g. line 63 after chemical, line 79 after interferon

and IFN, line 228 after Tamiflu, line 230 after Tamiflu, line 245 after IFN, etc.

We agree with the reviewer and this issue was due to a PDF conversion. In our original manuscript, the spaces and symbols are present and have been updated in the revised manuscript.

Line 37 it should be domain not domain

We are sorry, we do not understand what the reviewer means with this comment.

Line 44 The abbreviation TIMPS was already explained in previous paragraph

Agreed, it has been removed.

Line 49 it should be cancers not cancer

Changed.

Line 54 authors should provide the sequence of the mentioned peptide backbone 11

This sentence has been modified and a reference was added.

Lines 80-82 Sentence: This effect was associated with decreased serum level of MMP9 was found to be corelated with reduction in brain lesion activity as measured by gadolinium- enhancing lesion number change [28]. is not clear. Please rephrase.

Agreed. We have added more description and clarification of this clinical trial.

Line 85 should be 3’-phosphoinositide kinease-1 (missed dash)

Added.

Line 87 smaller than what ?

This has been clarified: “However, these clinical trials were performed only on a small number of patients and further studies on a larger number of patients are needed to further test their efficacy.”

Line 89 explanation for ECM abbreviation was given in introduction

This has been updated.

Lines 93-95 Authors should clarify or give examples which MMPs are considered as beneficial and which are not.

This has been updated and the reference “Dufour, A.; Overall, C.M. Missing the target: matrix metalloproteinase antitargets in inflammation and cancer. Trends Pharmacol. Sci. 2013, 34, 233–242. DOI: org/10.1016/j.tips.2013.02.004” is given where the full list of beneficial MMPs is presented.

Line 101 should be investigated not investigating

Agreed.

Lines 133-134 MMP9 or MMP-9 ? please unite

Line 134 has been updated to MMP9.

Line 153 powers should be in superscripts

Agreed and this was an PDF conversion issue.

Line 160 what compounds ? can authors provide structures? Line 169 can authors provide the structure of JNJ0966 ?

We have added a NEW figure (Figure 1a-g) showing the structure of these compounds and periostat.

Line 189 is SDS3 the antibody inhibitor ? Please clarify and rephrase the sentence.

Clarification of SDS3 as a monoclonal antibody has been added.

Line 196 should be Zn2+ (superscript)

This has been changed and was a PDF conversion issue.

Line 203 What cancer ? please specify.

Breast cancer. This has been added.

Line 206 Again, what cancer ?

Breast cancer. This has been added.

Lines 210-214 It also relates to breast cancer mentioned in line 209 ?

Yes both breast cancer and it has been modified in both sentences.

Line 287-288 Please develop this sentence “however, the broad-spectrum targeting has impeded their therapeutic applicability”

This has been modified to include broad-spectrum inhibitors targeting multiple MMPs.

Figure 1 needs more detailed explanation in caption.

Agreed and more information has been added.

If you need further information, please do not hesitate to contact us.

Sincerely,

Antoine Dufour, PhD

Assistant Professor

Cumming School of Medicine

University of Calgary

Calgary, AB, Canada
